# Better sleep, better life? testing the role of sleep on quality of life

**Michaela Kudrnáčová** [1]*, **Aleš Kudrnáč** [2]

**1** Department of Sociology, Faculty of Social Sciences, Charles University, Prague, Czechia, **2** Institute of Sociology of the Czech Academy of Sciences, Prague, Czechia

* 40672727@fsv.cuni.cz

## Abstract

Previous research has shown that sleep deprivation, low quality sleep or inconvenient sleeping times are associated with lower quality of life. However, research of the longitudinal effects of sleep on quality of life is scarce. Hence, we know very little about the long-term effect of changes in sleep duration, sleep quality and the time when individuals sleep on quality of life. Using longitudinal data from three waves of the Czech Household Panel Study (2018–2020) containing responses from up to 4,523 respondents in up to 2,155 households, the study examines the effect of changes in sleep duration, sleep quality and social jetlag on satisfaction with life, happiness, work stress, subjective health and wellbeing. Although sleep duration and timing are important, panel analyses reveal that sleep quality is the strongest predictor of all sleep variables in explaining both within-person and between-person differences in quality of life indicators.

**Data Availability Statement:** The fourth wave (2018) of the data is publicly available. The data are deposited to the public data depository (Czech Social Science Data Archive): http://nesstar.soc.cas.cz/webview/index.jsp?v=2&submode=abstract&study=http%3A%2F%2F147.231.52.

## Introduction

Previous research has shown that sleeping patterns are related to quality of life (QoL) and that key aspects are the time when individuals sleep, sleep duration and sleep quality. People who obtain sufficient high-quality sleep at proper times were found to have better general health [1] and overall quality of life [2]. By contrast, individuals who sleep too much [2] or sleep poorly [3] exhibit diminished quality of life. Despite previous research on QoL and sleep being substantial, they often lack in depth and scope and we know little about the effects of these three aspects of sleep on QoL and the development of their influence over time, which are significant considerations. Using three waves of the Czech Household Panel Study data, the present study contributes to the literature by examining the effect of sleep duration, sleep quality and social jetlag on five QoL indicators and exploring the trends in time.

## Quality of life definition

Originally, high QoL was perceived as a lack of stress, but the idea evolved into a multidimensional concept which emphasizes the subjectivity of experience, function and wellbeing and encompasses the physical, psychological and social domains of life [4]. QoL is an interplay between the perception of an internal state, such as the experience of happiness or feeling of

118%3A80%2Fobj%2FStudy%
2FCHPSEN401&mode=documentation&top=yes
CSDA is part of the Consortium of European Social
Science Data Archives (CESSDA). It provides open
access to social science data for non-commercial
use. The fifth and sixth wave of CHPS (2019-2020)
are strictly confidential due to the inclusion of
sensitive information and thus not publicly
available. However, the data could be shared upon
reasonable request on an individual basis. Also, to
comply with the PLOS ONE requirements, by the
time of manuscript publication, we will make sure
the data will be made publicly available at least to
the extend of the submitted article. Thus, we
decided to share the minimal dataset along with
syntax used in analyses in Supporting Information.

**Funding:** The work was supported by the grant
SVV at the Institute of Sociological Studies, Faculty
of Social Sciences, Charles University and by the
Czech Science Foundation (project 22-09220S).
The funders had no role in study design, data
collection and analysis, decision to publish, or
preparation of the manuscript

**Competing interests:** The authors have declared
that no competing interests exist.

good health or satisfaction, and external events in the surrounding environment, which may include family and career [5].

The model in the present study was built according to the theoretical model of QoL by Ventegodt et al. [6]. The model comprises various parameters grouped into three complementary categories, each being concerned with an aspect of good life: subjective, existential and objective. The above-mentioned authors incorporated notions of QoL into an *integrative quality-of-life* (IQOL) theory. We base our analysis on the subjective component of this all-embracing theory, which includes the following parameters: wellbeing, satisfaction with life, happiness and meaning in life (Fig 1).

These IQOL parameters are intertwined and crucial factors in describing QoL [4]. For instance, subjective wellbeing might be characterized as an emotional response and evaluation of satisfaction with life [7] which includes both cognitive judgments and affective reactions [4]. Since wellbeing captures a person's emotional state and touches on their mental state, our interpretation regards these states as complementary to subjective health, which more straightforwardly encompasses physical aspects. While happiness could be described as a person's current positive emotional condition [8], satisfaction with life represents a stable assessment of general feelings about life and indicates a long-term attitude [8]. Work also forms an important part of life, contributing to its meaning [6]. Although work can be exciting and satisfying, it may also be a cause of stress. Work stress refers to a negative psychological state which may involve numerous conditions in the working environment and consists of an interplay of cognitive, affective and physiological reactions functioning as stressors [9]. Stress causes the anatomic nervous system to release the hormone cortisol, which commonly aids in regulating sleep cycles. At elevated levels, however, cortisol results in sleep disturbances and insomnia [10]. Insufficient, excess, poor or otherwise impaired sleep, especially in the long-term, is concerning since it may result in severe physical, mental and social consequences in quality of life.

## Previous research

### Quality of life and its relationship to sleep

According to *Repair and Restoration theory* (RRT), sufficient sleep rewards us with restoration and repair that no other physiological process is able to achieve [11]. After a good night's sleep, individuals feel mentally sharp and rested. Research on body functioning also suggests that muscle repair, tissue growth and many other essential processes occur primarily during sleep [12], thereby affecting wellbeing and QoL. By contrast, insufficient sleep and accumulated sleep debt impairs mental function [13] and leads to health problems, including depression [14], obesity [15], diabetes and cardiovascular disease [16], increases the risk of cancer and reduces life expectancy [17]. IQOL and RRT theories and strong empirical evidence indicate that sleep affects QoL. Not only that sleep, in theory, restores the body and elevates the mind, studies have confirmed that sleep predicts quality of life, not the opposite [18, 19]. Previous research suggests three aspects of sleep are related to QoL: sleep duration, sleep quality and social jetlag.

**Sleep duration.** Sleep duration is a reliable predictor of wellbeing [18] and affects QoL. A systematic review and meta-analysis by Cappuccio et al. [20] found that both too short and too long periods of sleep lead to elevated mortality. There is, however, no agreement in the literature on what is normal, short or long sleep duration, each study used different cut-off points. This is also a reason why our study relates only to relative time spent sleeping (less or more hours in comparison to other respondents). A longitudinal study of 1,601 Swiss and Norwegian adolescents concluded that longer sleep duration is associated with higher levels of wellbeing [18]. In another study of adolescents (n = 4,582), shorter sleep duration was related to a

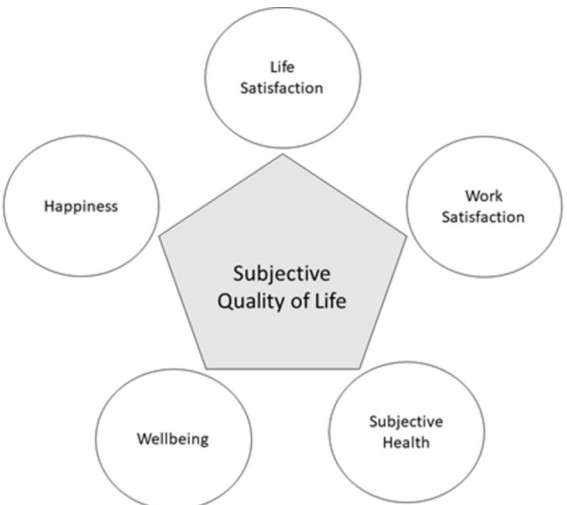

**Fig 1. Subjective quality of life according to integrative quality-of-life (IQOL) theory.** Note: Modified model of Søren Ventegodt et al. (2003:1032) integrative quality-of-life theory. The indicators for the five dimensions of quality of life refer to the indicators used in the Czech Household Panel Survey.

lower level of happiness [21]. Ness and Saksvik-Lehouillier [22] surveyed 474 Norwegian university students and concluded that longer average sleep duration is associated with greater life satisfaction.

However, some studies, such as a two-decade old experiment involving 75 university students who maintained sleep logs for three seven-day periods over three months and subsequently took part in a survey [23], claim that sleep quantity does not contribute to wellbeing. Two recent studies drawing on the German Socio-Economic Panel separately investigated sleep duration on workdays and weekends: Pagan [24] observed a sample of 105,340 individuals with disabilities for six years (2008–2013) and concluded that longer sleep duration on workdays increases life satisfaction. Piper [25] explored a sample of 68,782 individuals from the same panel data (2008–2012) and found that life satisfaction increases with longer sleep duration during workdays but not on weekends. In a study of 547 university students, Önder [26] found no correlation between sleep duration and happiness. However, the reliability of these conclusions is debatable since they were both based on small student samples, and the Turkish study involved mainly women (80.4%). Similarly, a longitudinal two-year study of 1139 Chinese university students indicated that sleep duration does not predict QoL [27]. Besides sleep duration, sleep quality is also related to wellbeing [1, 22, 23] and overall QoL [28–30].

**Sleep quality.** Although sleep quality is often considered affecting QoL more than sleep duration, they are not usually investigated together, the focus being solely on sleep quality. One notable example used a representative Austrian sample of 1,049 people and showed the significance of the relationship between sleep quality and QoL [30]. Research based on representative samples is scarce, and studies have principally involved student samples or patients.

Poorer sleep was found to be associated with adverse effects and significantly lower levels of happiness [21] and life satisfaction among Norwegian [22] and Korean students [19]. The above-mentioned small-scale experiment by [23] on college students in the US revealed no effect of sleep quantity on QoL but found sleep quality to be a strong and consistent long-term predictor of QoL. In an experiment on the interaction of sleep with campus residence and its effect on wellbeing, the authors of a Chinese study of university students concluded that overall

sleep quality deteriorated over time and that sleep had no significant effect on QoL [27]. Students are often used in experiments for their accessibility, but the general applicability of the results of studies on these samples is limited. Students are young, do not work in full-time employment, and their physiological and life characteristics differ from the general population. Other studies often use specific populations such as patients, the elderly or workers in certain heavy industries.

In a study of a specific adult and mostly male population of 145 patients diagnosed with schizophrenia, the conclusions resembled other reports in that poor sleepers tend to report lower QoL [29]. A longitudinal two-year Australian study of a sample of 93 adults with autism similarly concluded that poor sleep quality predicted poor QoL [31]. Jean-Louis et al. [1] collected sleep data on 273 adult San Diego residents (aged 40–64 years); their investigation revealed that self-perceived sleep quality is associated with wellbeing. Another cross-sectional study researched 435 female shift-working nurses in Taiwan and also concluded that poor sleep quality in the sample resulted in poorer life quality [28]. Disrupted sleep and therefore low-quality sleep, was also found to decrease QoL and increase work stress in a sample of 35,932 Korean workers [10].

**Social jetlag.** Previous studies have shown that sleep duration and sleep quality are crucial variables in predicting QoL. However, the time when individuals sleep is often overlooked. People must adjust the time when they sleep to social arrangements which do not often agree with their intrinsic preferences. This misalignment between our social and internal biological rhythms leads to social jetlag, which has previously been found to relate to QoL [32, 33]. The relationship between social jetlag and QoL is understudied, and the results of studies are inconsistent. Only two small-scale studies have been conducted on student samples, finding no link between social jetlag and QoL [26, 34]. Other studies have reported a negative correlation between social jetlag and QoL [35].

**Summary of previous research.** With the exception of some studies which used longitudinal data [18, 23–25, 31, 34], the majority of studies are cross-sectional [e.g., 22, 33, 36] and hence, a deficit in longitudinal panel studies exists. Only two studies exploring the effect of sleep variables on the quality of life are nationally representative [25, 30], while the remainder of studies were conducted on either a few dozen [23] or few hundred [e.g., 1, 22, 37] individuals and mainly examined specific populations, such as adolescents [19, 21, 34], university students [e.g., 19, 21, 22, 37], people with disabilities [24], people with autism [31] or patients diagnosed with schizophrenia [29]. Although Lau et al. [27] concluded that social jetlag predicted QoL, caution is required in interpreting their results. Their claim that social jetlag is reflected in perceived poorer sleep and impaired wellbeing is problematic, and their results are therefore debatable. The only accepted method of measuring social jetlag is the computation model developed by Roenneberg et al. [38]. Even though some studies have explored two aspects of sleep, for example, sleep duration and sleep quality [e.g., 21–23], or sleep duration and social jetlag [26], none have incorporated all three aspects (sleep duration, sleep quality, social jetlag), and hence, we have insufficient knowledge of the relative importance of the three most important sleep characteristics on QoL.

Based on the IQOL and RRT theories and the previous literature and considering the analytical methods allowing us to observe relative in-between and within differences, we formulated the following hypotheses on the role of sleep in QoL:

H1A. Individuals with on average longer sleep duration have higher levels of QoL than individuals with shorter sleep duration.

H1B. Increases in sleep duration over time are related to higher levels of QoL.

H2A. Individuals with on average higher sleep quality have higher levels of QoL than individuals with lower sleep quality

H2B. Improvements in sleep quality over time are related to higher levels of QoL.

H3A. Individuals with on average lower social jetlag levels have higher levels of QoL than individuals with higher social jetlag.

H3B. Decreases in social jetlag over time are related to higher levels of QoL.

## Data and methods

### Study design and participants

The analyses used data from the Czech Household Panel Survey (CHPS) *which focuses on mapping the living conditions and describing the dynamics of change among both Czech households and individuals in the long-term perspective* [39]. These data were collected annually from 2015 until 2020, typically between the end of June and the end of October. A two-stage stratified random sampling method was applied and the design effects were further mitigated by the use of a large number of small primary sampling units. The original sampling frame from the very first data collection consisted of the Register of Census Districts and Buildings which had been transformed into an address database. Since the target population was the non-institutionalized population of the Czech Republic, all members of the sampled households were interviewed. In each of the following waves, the same members of the same households participating in the preceding wave were approached (e.g., in wave three in 2017, only participants from wave two were approached). The data are nationally representative of the adult population in CR. The retention rate of households between the first and sixth waves of data collection was 21.6% on average, and the retention rate of individuals was 20.6%. All information regarding data collection including survey design is available in the Czech Social Science Data Archive [39].

A total of 5,132 paper-and-pencil self-administered questionnaires (SAQ) incorporating the key variables were collected from Czech adults in 2018, 2,046 in 2019, and 2,161 questionnaires in 2020. The final dataset contained responses from up to 4,523 respondents in up to 2,155 households. The significant drop in the sample between 2018 and 2019 was caused by the blood draw requirement. Sleep variables were included into the questionnaires during the waves 4–6 (2018–2020) due to the collaboration between Institute of Sociology and Institute of Physiology of the Czech Academy of Sciences at that time. They were measured according to the Munich Chronotype Questionnaire (MCTQ): some were measured, and some were computed (for more information on used variables, see the section Measures down below). Written informed consent was obtained from all respondents. The study followed the principles of the Declaration of Helsinki and was approved by Ethics Committee of the Institute for Clinical and Experimental Medicine and Thomayer Hospital in Prague (study number G-16–05–02).

The data from the CHPS are widely used by researchers for secondary data analysis: for instance, studies are focusing on certain aspects of sleep, specifically chronotype assessment [40] and social jetlag in the work-family context [41], other studies explore the division of housework and relative resources [42] partnership trajectories [43], mechanisms of the reproduction of homeownership [44], voter turnout [45].

### Measures

We investigated the effect of sleep on the five dependent variables which describe QoL: life satisfaction, wellbeing, happiness, subjective health and work stress. At all points in time, **life**

**satisfaction** was measured with responses to the question "All things considered, how satisfied are you with your life as a whole?" The response options were scaled from zero to ten, zero indicating "extremely dissatisfied" and ten indicating "extremely satisfied". Many other studies have used the same items to measure life satisfaction [e.g., 46, 47].

**Wellbeing** was calculated as an average of three items to measure how often over the last two weeks respondents "have been cheerful and in good spirits"; "have felt calm and relaxed"; "have been active and vigorous". The six response options with scores from one to six were "at no time", "some of the time", "less than half of the time", "more than half of the time", "most of the time", "all of the time". The resultant reliability estimates are acceptable ($\alpha_{t1}$ = .811; $\alpha_{t2}$ = .828 $\alpha_{t3}$ = .830; $\alpha_{t4}$ = .841 $\alpha_{t5}$ = .825). The scale was computed as a sum of means also ranging from one to six. The same items were measured during two out of three analysed years of data collection in 2018 and 2019 and have also been used to measure wellbeing in other studies [e.g., 48].

Perceived **happiness** was measured with the question "Taking all things together, how happy would you say you are?". The respondents were asked to answer on a scale of zero to ten, zero indicating "extremely unhappy" and ten indicating "extremely happy". The same items were measured during two out of three analysed years of data collection in 2018 and 2019 and have also been used to measure happiness in other studies [e.g., 47, 49].

Respondents rated their **subjective health** according to the question "In general, would you say your health is. . .?" on a five-point scale of "poor", "fair", "good", "very good" and "excellent". The same items were measured during two out of three analysed years of data collection in 2018 and 2019 and have also been used to measure subjective health in other studies [e.g., 47, 49].

The respondents' perceived **work stress** was calculated according to the proportion of affirmative answers to the question "Have the following circumstances in your current job caused you excess worry or stress in the past 12 months?" according to the following items: "threat of layoffs or losing the job"; "workplace safety, accidents, or injuries on the job"; too many demands or too many working hours at work." The response options were "yes" or "no". The same items were measured during one wave (2018) during the reference period. The questions are proxies inspired by the European Working Conditions Surveys (EWCS).

In addition to the dependent variables, three facets of sleep were measured. Specifically, we assessed the average sleep duration, perceived sleep quality, and social jetlag. **Sleep duration** was calculated as the average of answers to questions regarding the time when respondents usually fell asleep and woke up on free days and when they usually fell asleep and woke up on workdays. The same items were measured during the complete analyzed period: wave 4 (2018), wave 5 (2019) and wave 6 (2020) of data collection and have also been used to measure sleep quality in other studies [e.g., 33, 40].

Perceived **quality of sleep** was measured with the question "How would you rate the quality of your sleep?" according to a four-point Likert scale for the response options "very bad", "bad", "good" and "very good". The same items were measured during the complete analyzed period: wave 4 (2018), wave 5 (2019) and wave 6 (2020) of data collection and have also been used to measure sleep quality in other studies [e.g., 50, 51].

**Social jetlag** was calculated according to a MCTQ [52] as the difference between the mid-sleep time on free days and workdays. The resultant values were converted into numeric variables which represented the hours. The results were interpreted as follows: zero indicated no sleep debt during workdays or free days, and any values above zero indicated an accumulation of sleep debt. The same items were available during the complete analyzed period: wave 4 (2018), wave 5 (2019) and wave 6 (2020) of data collection and have also been used to measure social jetlag in other studies [e.g., 14, 33, 34].

Data on age, gender, highest level of education attained (basic and secondary vocational, secondary with maturita, tertiary education), net household income (Net household income is stated in Czech Crowns (CZK). For illustration, according to the European Union–Statistics on Income and Living Conditions (EU-SILC), the average monthly net income of a Czech household reached CZK 17.5 thousand per person in 2019 [53]. The net household income categories used in this article can be roughly converted to USD as it follows: 1 = up to 918 USD, 2 = 918 to 1,197 USD, 3 = 1,198 to 1,396 USD, 4 = 1,397 to 1,596 USD , 5 = 1,597 to 2,993 USD, 6 = more than 2,994 USD.) (1 = up to CZK 22,999, 2 = CZK 23,000 to 29,999, 3 = CZK 30,000 to 34,999, 4 = CZK 35,000 to 39,999, 5 = CZK 50,000 to 74,999, 6 = more than CZK 75,000), number of children below the age of five in the household, and economic status were also collected and controlled for (45.80% employed, 6.20% self-employed, 2.90% unemployed, 8.88% students, 33.07% retired, and 3.14% on maternity leave). The descriptive statistics for all variables used in our analyses is reported in Table 1.

## Statistical analysis strategy

To test our hypotheses on the effects of the three measured aspects of sleep on life satisfaction, wellbeing, happiness, subjective health and work stress, we analysed the CHPS data according to mixed, multilevel repeated measurement models with random intercepts for individuals, households and a random slope for time. To examine whether sleep quality, sleep duration and social jetlag would predict between-person and within-person changes in the dependent variables, we constructed three-level hierarchical models with time nested within both individuals and households. The variables at the within-person level were person-mean-centred and constituted a measurement of the degree to which an individual's characteristics changed over time. The variables at the between-person level were grand-mean-centred and tested whether and how much individuals differed from each other.

We started with null models which incorporated the dependent variables without predictors to capture the variance of the dependent variables (S1 Table). Next, we examined the longitudinal effects of the three tested facets of sleeping hygiene on the five measures of QoL by adding sleep duration, sleep quality and social jetlag variables and interaction terms for time and sleeping variables (Models 1A–5A). In the final step, Models 1B–5B decomposed the effects of sleeping on within-person and between-person effects. We then evaluated the model

**Table 1. Descriptive statistics of used variables.**

| | Number of respondents | | Mean | Std. Dev. | Min | Max |
|---|---|---|---|---|---|---|
| **Gender** (2018, 2019, 2020) | 4,523 | | 1.58 | 0.49 | 1 | 2 |
| **Education** (2018, 2019, 2020) | 4,523 | | 1.95 | 0.76 | 1 | 3 |
| **Age** (2018, 2019, 2020) | 4,523 | | 51.93 | 16.766 | 18 | 96 |
| **Household income** (2018, 2019, 2020) | 4,523 | | 3.79 | 1.78 | 1 | 6 |
| **Economic status** (2018, 2019, 2020) | 4,523 | | 1.79 | 1.92 | 0 | 5 |
| **Social jetlag** (2018, 2019, 2020) | 4,523 | | 0.87 | 0.87 | 0 | 5.83 |
| **Sleep duration** (2018, 2019, 2020) | 4,523 | | 7.48 | 1.12 | 3.5 | 13.48 |
| **Quality of sleep** (2018, 2019, 2020) | 4,523 | | 3.00 | 0.68 | 1 | 4 |
| **Children below the age of 5** (2018,2020) | 4,523 | | 0.18 | 0.47 | 0 | 2 |
| **Life satisfaction** (2018, 2019, 2020) | 4,523 | | 7.47 | 1.79 | 0 | 10 |
| **Wellbeing** (2018, 2019) | 3,850 | | 4.08 | 0.92 | 1 | 6 |
| **Subjective health** (2018, 2019) | 3,867 | | 3.12 | 1.00 | 1 | 5 |
| **Working stress** (2018, 2020) | 2,097 | | 0.19 | 0.24 | 0 | 1 |
| **Happiness** (2018, 2019) | 3,857 | | 7.34 | 1.77 | 0 | 10 |

fits according to the general principle that models with lower deviance and AIC values than the null model are considered better fitting models [54].

## Results

Initially, we built models without predictors to examine the variance in all five of the measured aspects of quality of life. These null models (S1 Table) showed 47% variance in life satisfaction between individuals and 23% variance between households, 56% variance in wellbeing between individuals and 20% variance between households, 74% variance in subjective health between individuals and 26% variance between households, 51% variance in working stress between individuals and 12% variance between households, 56% variance in happiness between individuals and 23% variance between households.

### Do changes in sleep affect the quality of life over time?

To test the effect of sleep over time, we added sleep duration, sleep quality, social jetlag, control variables, the fixed effect of time and interaction term for time, and each of the three variables which capture sleeping (Table 2: Models 1A–5A). The variables improved model fit in all models (life satisfaction: Δ-2LL = 247.68 (16), p < .001; ΔAIC = 215.68; wellbeing: Δ-2LL = 404.70 (16), p < .001; ΔAIC = 372.70; health: Δ-2LL = 1307.65 (16), p < .001; ΔAIC = 1275.65; work stress: Δ-2LL = 106.62 (16), p < .001; ΔAIC = 74.62; happiness: Δ-2LL = 296.54 (16), p < .001; ΔAIC = 262.54).

The interaction of sleep duration and the time variable was a positive and statistically significant predictor of wellbeing (B = .092, p < .001), subjective health (B = .060, p = .005), and happiness (B = .148, p = .001). The effect of the interaction term was not a statistically predictor in the model for life satisfaction (B = −.019, p = .497) or work stress (B = −.003, p = .575).

The interaction of sleep quality and the time variable was not a statistically significant predictor of any of the tested dependent variables (subjective health: B = −.029, p = .391; happiness: B = −.115, p = .110; life satisfaction: B = .050, p = .268; wellbeing: B = −.017, p = .652; work stress: B = .006, p = .562).

The interaction of social jetlag and the time variable was a negative and statistically significant predictor of wellbeing (B = −.062, p = .042), but not a statistically significant predictor in the models for subjective health (B = −.041, p = .136), happiness (B = .022, p = .700), life satisfaction (B = −.016, p = .678) or work stress (B = −.005, p = .536).

A graphical representation of the calculated marginal effects highlighted the differences in QoL between individuals who slept fewer or more hours on average (Fig 2), perceived their sleep to be worse or better quality (Fig 3), and suffered from less or more social jetlag (Fig 4), whereas other variables remained at their mean values.

### Does sleep predict within-person and between-person changes in quality of life?

Further examination of the longitudinal effect of sleep on quality of life in Models 1B–5B (Table 3) distinguish the discussed between-person and within-person effects. Separation of the between-person and within-person effects improved model fit in the models for predicting life satisfaction (Δ-2LL = 37.44 (1), p < .001; ΔAIC = 37.44), wellbeing (Δ-2LL = 15.25 (41), p < .001; ΔAIC = 13.25), health (Δ-2LL = 50.66 (1), p < .001; ΔAIC = 50.66) and happiness (Δ-2LL = 16.69 (1), p < .001; ΔAIC = 18.69) but did not show any statistically significant improvement in model fit for work stress (Δ-2LL = 3.184 (1), p < .074; ΔAIC = 3.18) over Models 1A–5A.

**Table 2. Sleeping habits and quality of life, linear mixed models with repeated measurements.**

| | | Model 1A | | Model 2A | | Model 3A | | Model 4A | | Model 5A | |
|---|---|---|---|---|---|---|---|---|---|---|---|
| | | Life satisfaction | | Wellbeing | | Subjective health | | Work stress | | Happiness | |
| **Interaction terms** | | | | | | | | | | | |
| | Sleep duration*time | −.02 | (−.07 - .04) | .09*** | (.04 - .14) | .06** | (.02 - .10) | < .01 | (−.02 - .01) | .15*** | (.06 - .24) |
| | Sleep quality*time | .05 | (−.04 - .14) | −.02 | (−.09 - .06) | −.03 | (−.10 - .04) | .01 | (−.01 - .02) | −.11 | (−.26 - .03) |
| | Social jetlag*time | −.02 | (−.09 - .06) | −.06* | (−.12 - −.01) | −.04 | (−.09 - .01) | < .01 | (−.02 - .01) | .02 | (−.09 - .13) |
| **Sleep variables** | | | | | | | | | | | |
| | Sleep duration | .06 | (−.14 - .26) | −.30*** | (−.46 - −.15) | −.23** | (−.37 - −.09) | < .01 | (−.04 - .05) | −.56*** | (−.85 - −.27) |
| | Sleep quality | .32 | (−.01 - .65) | .45*** | (.19 - .70) | .47*** | (.24 - .69) | −.06 | (−.12 - .01) | .99*** | (.52–1.47) |
| | Social jetlag | .02 | (−.25 - .30) | .20 | (−.01 - .40) | .13 | (−.05 - .30) | .03 | (−.02 - .09) | −.12 | (−.49 - .26) |
| **Socio-demographic variables** | | | | | | | | | | | |
| | Time | .01 | (−.46 - .47) | −.53** | (−.93 - −.13) | −.33 | (−.68 - .03) | < .01 | (−.10 - .10) | −.69 | (−1.43 - .05) |
| | Gender (ref. cat. male) | .08 | (−.02 - .19) | .07* | (.02 - .13) | .09** | (.03 - .14) | −.01 | (−.03 - .01) | .14* | (.03 - .25) |
| | Education–secondary with maturita | .10 | (−.03 - .23) | −.03 | (−.10 - .04) | .14*** | (.07 - .21) | −.03 | (−.05 - .00) | .08 | (−.05 - .22) |
| | Education–tertiary | .09 | (−.06 - .25) | −.02 | (−.10 - .06) | .26*** | (.18 - .34) | −.04** | (−.07 - −.01) | .15 | (−.01 - .31) |
| | Age | .01** | (.00 - .02) | < .01 | (−.01 - .00) | −.02*** | (−.02 - −.02) | <−.01** | (−.01 - −.01) | .01** | (.00 - .01) |
| | Household income | .14*** | (.10 - .17) | .01 | (−.01 - .03) | .04*** | (.02 - .06) | < .01 | (−.01 - .00) | .12*** | (.08 - .16) |
| | Economic status (ref. cat. employed) | | | | | | | | | | |
| | Self-employed | .06 | (−.17 - .29) | −.09 | (−.21 - .03) | .05 | (−.07 - .17) | .02 | (−.02 - .06) | .02 | (−.22 - .26) |
| | Unemployed | −.31 | (−.65 - .04) | −.22* | (−.40 - −.04) | −.27** | (−.44 - −.09) | −.04 | (−.12 - .03) | −.12 | (−.47 - .23) |
| | Student | .58*** | (.30 - .87) | .17* | (.02 - .32) | .24** | (.10 - .38) | −.10*** | (−.16 - −.04) | .49** | (.19 - .78) |
| | Retired | .08 | (−.12 - .29) | < .01 | (−.11 - .11) | −.20*** | (−.30 - −.09) | −.10*** | (−.15 - −.05) | .22* | (.00 - .43) |
| | Maternity leave | .27 | (−.05 - .58) | −.04 | (−.20 - .13) | −.03 | (−.19 - .14) | −.10** | (−.17 - −.03) | .02 | (−.31 - .34) |
| | Number of children below the age of 5 | | | | | | | | | | |
| | One child | .18 | (−.03 - .39) | .02 | (−.09 - .12) | .08 | (−.02 - .17) | < .01 | (−.02 - .04) | .38*** | (.16 - .59) |
| | Two or more children | .41* | (.07 - .76) | .06 | (−.12 - .24) | .16 | (−.00 - .33) | <-0.01 | (−.07 - .05) | .38* | (.03 - .74) |
| **Constant** | | 4.82*** | (3.06–6.57) | 4.76*** | (3.41–6.11) | 3.94*** | (2.74–5.14) | .45* | (.09 - .82) | 7.15*** | (4.63–9.67) |
| | Observations | 4,523 | | 3,850 | | 3,867 | | 2,097 | | 3,857 | |
| | Households | 2,155 | | 2,100 | | 2,105 | | 1,305 | | 2,101 | |
| | AIC | 17502 | | 9551 | | 9013 | | -60 | | 14623 | |
| | BIC | 17662 | | 9701 | | 9169 | | 81 | | 14779 | |
| | ICC households | 15% | | 5% | | 7% | | 3% | | 5% | |
| | ICC individuals | 69% | | 94% | | 82% | | 95% | | 96% | |
| | ll | -8726 | | -4752 | | -4482 | | 55.31 | | -7287 | |

Note: *** p<0.001,

** p<0.01,

* p<0.05, 95% CI in parentheses

The effects of sleep duration on subjective health (B = −.045, p = .001) and happiness (B = −.084, p = .003) were statistically significant at the between-person level. Sleep duration was

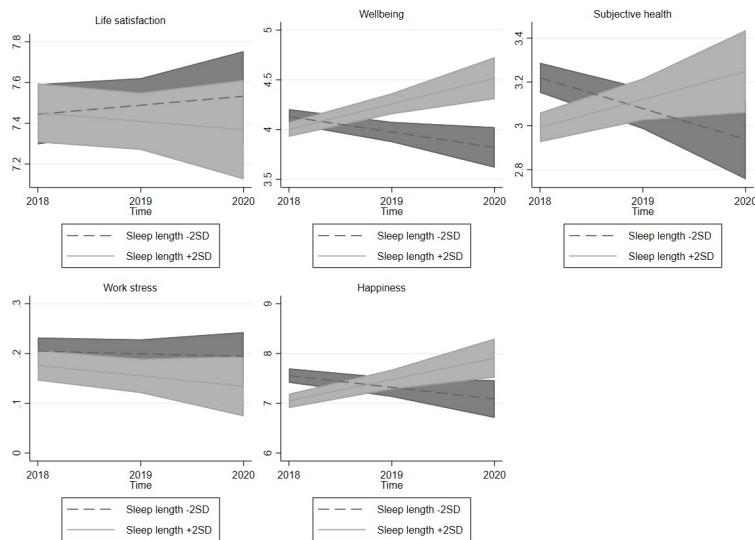

**Fig 2. Sleep duration and quality of life at the individual level in time.** Note: 95% confidence intervals.

not a statistically significant predictor of life satisfaction (B = −.021, p = .436), wellbeing (B = −.013, p = .364) or work stress (B = −.005, p = .424) at the between-person level. At the within-person level, the effects of sleep duration were not a statistically significant predictor of happiness (B = −.044, p = .405), wellbeing (B = .028, p = .335), subjective health (B = −.020, p = .412), work stress (B = −.022, p = .110) or life satisfaction (B = −.007, p = .890).

The effects of sleep quality on life satisfaction (B = .653, p < .001), wellbeing (B = .463, p < .001), work stress (B = −.043, p < .001) subjective health (B = .468, p < .001) and happiness (B = .742, p < .001) were statistically significant at the between-person level. At the within-person level, the effects of sleep quality were a statistically significant predictor of life satisfaction (B = .149, p = .036), wellbeing (B = .183, p < .001), subjective health (B = .143, p < .001) and happiness (B = .283, p < .001), but not of work stress (B = −.009, p = .612).

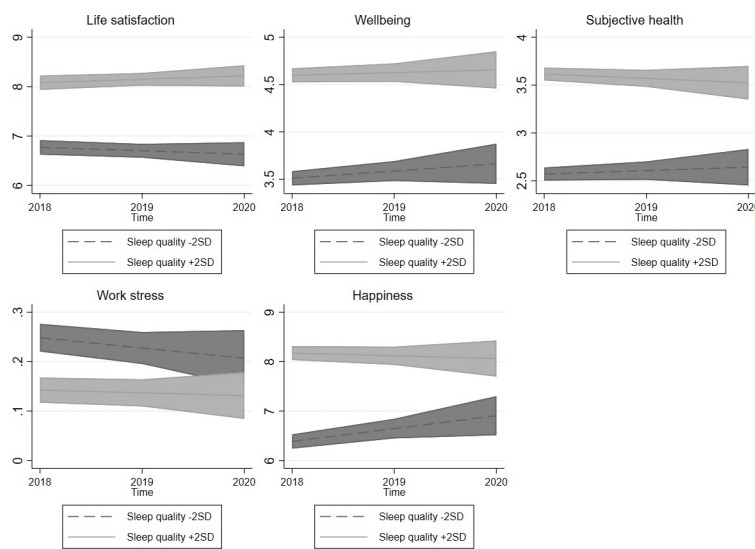

**Fig 3. Sleep quality and quality of life at the individual level in time.** Note: 95% confidence intervals.

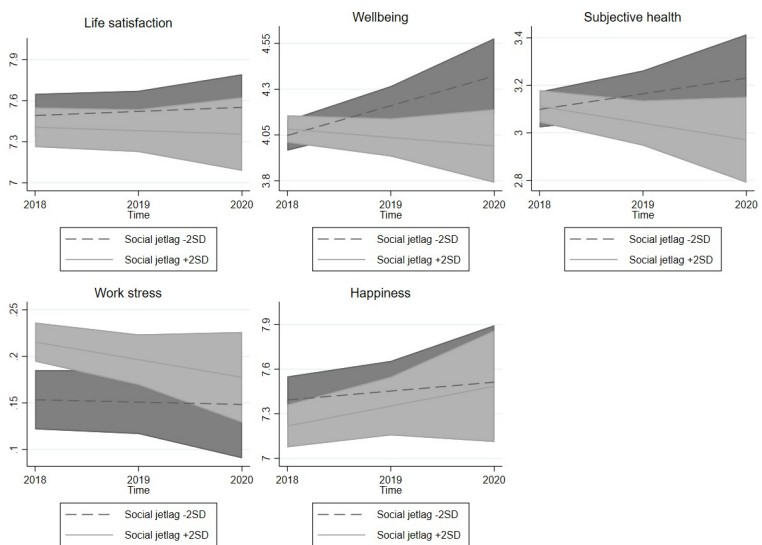

**Fig 4. Social jetlag and quality of life at the individual level in time.** Note: 95% confidence intervals.

The effects of social jetlag on life satisfaction (B = −.086, p = .040) and work stress (B = .017, p = .020) were statistically significant at the between-person level. Social jetlag was not a statistically significant predictor of happiness (B = −.083, p = .052), wellbeing (B = −.019, p = .378) or health (B = −.032, p = .125) at the between-person level. At within-person level, the effects of social jetlag on life satisfaction (B = .105, p = .136), wellbeing (B = .032, p = .433), work stress (B = .017, p = .296), happiness (B = .068, p = .352) and health (B = .059, p = .087) were not statistically significant.

## Discussion

The Czech Republic (CR) is comparable to other European countries in standard of living. The CR is on average commensurable with other European countries in life expectancy and economic activity [55] and self-perceived health [56]. While the life satisfaction score in the CR is very close to the European average, Czechs are slightly less happy, their happiness score being comparable to European countries such as Portugal, Italy and Greece [57]. The average sleep duration in the CR is 7.5 hours (see Data and methods section), which is similar to other European countries such as Belgium, France, Hungary, the Netherlands and the United Kingdom [58]. The proportion of Czechs (31%) with social jetlag is also comparable to the European average [15]. However, although Czechs report around 49 minutes of social jetlag (see Data and methods section), Spaniards and Germans report longer times [59]. The source of this difference is unclear, but it is probably because the samples are non-representative. It may be also the result of distinct cultural and environmental contexts or locations. In summary, the CR represents a case study of a population with living standards, QoL and sleep patterns are comparable to other European countries. The findings of the present study can therefore be reasonably generalized to other countries.

Building on IQOL theory and previous studies, the present study expands on the relationship between QoL and sleep. It contributes to the existing literature by examining the main areas of life and sleep from representative panel data to form a better understanding of how sleep and QoL are intertwined and the development of their relationship over time. The results of this study do not support the hypothesis (H1B) that QoL increases when people change their sleeping habits to spend more time sleeping. However, the results agree with previous

**Table 3. Sleeping habits and quality of life, linear mixed models with repeated measurements.**

| | | Model 1B | | Model 2B | | Model 3B | | Model 4B | | Model 5B | |
|---|---|---|---|---|---|---|---|---|---|---|---|
| | | Life satisfaction | | Wellbeing | | Subjective health | | Work stress | | Happiness | |
| **Sleep variables— between person level** | | | | | | | | | | | |
| | Sleep duration | −.02 | (−.08 - .03) | −.01 | (−.04 - .02) | −.04** | (−.07 - −.02) | < .01 | (−.02 - .01) | −.08** | (−.14 - −.03) |
| | Sleep quality | .65*** | (.56 - .74) | .46*** | (.42 - .51) | .47*** | (.42 - .51) | −.04*** | (−.06 - −.03) | .74*** | (.65 - .84) |
| | Social jetlag | −.09* | (−.17 - −.00) | −.02 | (−.06 - .02) | −.03 | (−.07 - .01) | .02* | (.00 - .03) | −.08 | (−.17 - .00) |
| **Sleep variables— within person level** | | | | | | | | | | | |
| | Sleep duration | −.01 | (−.11 - .09) | .03 | (−.03 - .08) | −.02 | (−.07 - .03) | −.02 | (−.05 - .01) | −.04 | (−.15 - .06) |
| | Sleep quality | .15* | (.01 - .29) | .18*** | (.10 - .26) | .14*** | (.07 - .21) | −.01 | (−.04 - .03) | .28*** | (.14 - .43) |
| | Social jetlag | .11 | (−.03 - .24) | .03 | (−.05 - .11) | .06 | (−.01 - .13) | .02 | (−.01 - .05) | .07 | (−.08 - .21) |
| **Socio-demographic variables** | | | | | | | | | | | |
| | Time | .03 | (−.03 - .09) | .05* | (.00 - .10) | < .01 | (−.05 - .04) | −.01* | (−.02 - −.00) | .09* | (.00 - .18) |
| | Gender (ref. cat. male) | .10 | (−.00 - .21) | .08** | (.03 - .14) | .10*** | (.04 - .15) | −.01 | (−.03 - .01) | .15** | (.04 - .26) |
| | Education–secondary with maturita | .09 | (−.04 - .22) | −.02 | (−.09 - .04) | .14*** | (.07 - .20) | −.03 | (−.05 - .00) | .08 | (−.06 - .22) |
| | Education–tertiary | .08 | (−.08 - .23) | −.02 | (−.11 - .06) | .25*** | (.17 - .33) | −.04* | (−.07 - −.01) | .14 | (−.02 - .30) |
| | Age | .01** | (.00 - .02) | < .01 | (−.01 - .00) | −.02*** | (−.02 - −.02) | <−.01** | (−.01 - −.01) | .01* | (.00 - .01) |
| | Household income | .13*** | (.10 - .17) | .01 | (−.01 - .03) | .04*** | (.02 - .05) | < .01 | (−.01 - .00) | .12*** | (.08 - .16) |
| | Economic status (ref. cat. employed) | | | | | | | | | | |
| | Self-employed | .05 | (−.18 - .28) | −.09 | (−.21 - .03) | .05 | (−.07 - .16) | .02 | (−.02 - .05) | .01 | (−.23 - .25) |
| | Unemployed | −.31 | (−.65 - .03) | −.21* | (−.39 - −.03) | −.27** | (−.44 - −.09) | −.05 | (−.12 - .03) | −.12 | (−.47 - .23) |
| | Student | .55*** | (.26 - .83) | .16* | (.01 - .31) | .22** | (.08 - .37) | −.10*** | (−.16 - −.04) | .45** | (.16 - .75) |
| | Retired | .06 | (−.15 - .26) | < .01 | (−.11 - .11) | −.20*** | (−.31 - −.10) | −.10*** | (−.15 - −.05) | .20 | (−.01 - .42) |
| | Maternity leave | .25 | (−.07 - .57) | −.03 | (−.20 - .13) | −.03 | (−.20 - .13) | −.10** | (−.17 - −.03) | .01 | (−.32 - .34) |
| | Number of children below the age of 5 | | | | | | | | | | |
| | One child | .18 | (−.03 - .40) | .02 | (−.09 - .13) | .08 | (−.02 - .18) | .01 | (−.02 - .04) | .38*** | (.17 - .59) |
| | Two or more children | .41* | (.07 - .75) | .08 | (−.11 - .26) | .18* | (.01 - .34) | −.01 | (−.07 - .05) | .40* | (.05 - .76) |
| **Constant** | | 4.47*** | (3.82–5.12) | 2.74*** | (2.39–3.10) | 2.71*** | (2.38–3.04) | .50*** | (.37 - .63) | 4.36*** | (3.67–5.05) |
| | Observations | 4,523 | | 3,850 | | 3,867 | | 2,097 | | 3,857 | |
| | Households | 2,155 | | 2,100 | | 2,105 | | 1,305 | | 2,101 | |
| | AIC | 17464 | | 9538 | | 8962 | | -64 | | 14604 | |
| | BIC | 17625 | | 9694 | | 9118 | | 77 | | 14754 | |
| | ICC households | 14% | | 7% | | 5% | | 3% | | 10% | |
| | ICC individuals | 70% | | 87% | | 85% | | 95% | | 88% | |
| | ll | -8707 | | -4744 | | -4456 | | 56.91 | | -7278 | |

Note: *** p<0.001,

** p<0.01,

* p<0.05, 95% CI in parentheses

studies which report a relationship between sleep duration and QoL [18, 21] from results which show differences between people in their perceived health and happiness according to

the number of hours they spend sleeping (H1A). Individuals who spent more time sleeping also reported worse subjective health and lower levels of happiness. The negative association between subjective health and sleep duration may be a result of long-term stress or mental illness which have affected their sleeping habits since previous studies have shown that individuals with poor mental health and depressive symptoms report sleeping issues and also longer sleep duration [60]. The negative association between sleep duration and QoL agrees with previous findings [2, 18, 21].

In accordance with our hypotheses (H2A and H2B) and previous studies, sleep quality was found to be a robust and reliable predictor of QoL [1, 29, 30]. Our analyses show individuals who experience higher quality sleep also have greater satisfaction with life, more wellbeing, feel healthier, perceive less work stress and are happier (H2B). With changes over time, a positive association between improvement in quality of sleep and increase in life satisfaction, wellbeing, subjective health, and happiness is evident (H2A). The overall positive effect of change in sleep quality on QoL agrees with previous research [1, 10, 28, 30]. The only indicator not associated with a change in sleep quality is work stress, perhaps due to the complexity of the link between these indicators. A mediator variable which also captures emotional aspects, such as workplace relationships, might be missing [21].

These results also contribute to the ongoing debate regarding the ambiguous consequences of social jetlag on our lives. Our results agree with Jankowski [34] and Önder [26] are contrary to Chang and Jang [35]. Our hypothesis (H3A) that individuals with a higher level of social jetlag are less satisfied with life and experience a higher level of work stress than others was only partially confirmed. Our findings do not suggest any association between social jetlag and wellbeing, subjective health or happiness. Furthermore, a change in social jetlag has no effect on any measured QoL aspect (H2B). This may be due to social jetlag being relatively stable, as it is likely to change only as a consequence of a relatively major life change (new job, birth of a child) which results in a new sleep schedule. Therefore, individuals with less sleep debt experience a minor increase in various aspects of QoL, but individuals with more social jetlag stagnate, apart from experiencing a decrease in work stress. Since these changes are not very frequent, social jetlag has a low variation over time, leading to the absence of a longitudinal effect, except in work stress, which is most likely related to changes in employment arrangements.

The results of the present study are consistent with previous studies [1, 22, 28] and suggest a strong relationship between sleep quality and QoL and a rather limited effect of sleep duration or social jetlag on QoL. A comparison of the respondents' sleep quality indicated a slight improvement in happiness in those who experienced poorer sleep during the last wave (2020) of data collection. This may have been caused by an overall increase in sleep quality triggered by social lockdowns designed to suppress Covid-19. Poor sleepers also indicated a small decline in work stress, perhaps because of more flexible working arrangements experienced early during the Covid-19 pandemic. Longitudinal effects nonetheless remained stable over the previous three years, as we presumed.

The results of this longitudinal study provide an important insight into people's lifestyles. Despite people having different sleep requirements, the results suggest that both average sleep duration and social jetlag remain moderately stable over time. Sleep quality is also a valuable subjective measure related to other factors which encompass several important areas of life, such as mental and physical health, emotional wellbeing, cognitive functioning and feeling of safety.

## Limitations

The strengths of our study are longitudinal design, differentiation of between-person and within-person effects and the advantage of a representative dataset which enabled the

incorporation of all three aspects of sleep (quantity, quality, social jetlag) into a single model. This is also the first study which has tested the longitudinal effect of social jetlag on QoL. Admittedly, the study also has limitations. First, the period of measurement is relatively too brief to allow stronger claims regarding the longitudinal effect of sleep. Second, all the results are correlational. Using panel data does not qualify for asserting causal claims, and therefore it is not possible to state, for example, whether people feel less healthy because of low quality sleep or whether low-quality sleep leads to poorer health. Third, even though the CR is comparable to other European countries in living standard and sleeping habits, this is a case study of a single country. Having the opportunity to test our findings in other countries would be a great venue for future research. Fourth, the sleep indicators are self-reported and therefore have limitations despite self-reported measures being similarly reliable predictors [61]. Ideally, the measures would be collected in a medical lab or via mobile devices to aid in cross validating our results with more objective methods of measurement. Fifth, even though data were collected on regular days, the final wave partially captured the experience of the pandemic in the spring of 2020, and this study, therefore, might not be representative of the behavior under normal circumstances. However, data collection occurred during periods of eased restrictions and likely did not affect the generalizability of the results.

## Conclusion

The present study delivers a comprehensive analysis built on previous studies to extend knowledge on the role of sleep in life. In measuring three distinct facets of sleep in a single longitudinal model, sleep quality was found to be the most influential factor affecting the five aspects of QoL (wellbeing, life satisfaction, subjective health, work stress and happiness). Individuals who experienced more quality sleep also reported better QoL. Improvement of sleep quality over time is also related to improvements in QoL. Sleep duration and social jetlag are also somewhat related to QoL, but in contrast to sleep quality, these factors do not appear significant. The study suggests, with the exception of extremes, that sleep duration alongside the differences in sleep habits on workdays and free days is not as important to QoL as what is considered a good night's sleep. Sleep is vital to our functioning. Changes in lifestyle and psychological challenges which have either emerged or been amplified under the currently ongoing pandemic have undoubtedly affected sleeping habits. That topic, preferably in a study involving multiple points over time for a long-term comparison and sleep at non-standard times such as Covid-19 pandemic, will be the focus of future research.

## Supporting information

**S1 File. Replication dataset and replication syntax.**
(ZIP)

**S1 Table. Null models of sleeping habits and quality of life, linear mixed models with repeated measurements.** Note: *** p<0.001, ** p<0.01, * p<0.05, 95% CI in parentheses.
(PDF)

## Acknowledgments

This publication used data acquired through the data services of the Czech Social Science Data Archive (ČSDA). The CSDA research infrastructure project is supported by the Ministry of Education, Youth and Sports within the framework of grant LM2018135. Data collection was funded by the grant "Cumulative effects of social disadvantage on health and the quality of life" (reg. No. TL02000190), financed by the Technology Agency of the Czech Republic. These are

not direct fundings of this study, but recognition of the projects within which were the data collected and made available to the public.

## Author Contributions

**Conceptualization:** Michaela Kudrnáčová.

**Data curation:** Michaela Kudrnáčová.

**Formal analysis:** Aleš Kudrnáč.

**Funding acquisition:** Michaela Kudrnáčová, Aleš Kudrnáč.

**Methodology:** Aleš Kudrnáč.

**Project administration:** Michaela Kudrnáčová.

**Supervision:** Aleš Kudrnáč.

**Validation:** Michaela Kudrnáčová.

**Visualization:** Michaela Kudrnáčová, Aleš Kudrnáč.

**Writing – original draft:** Michaela Kudrnáčová, Aleš Kudrnáč.

**Writing – review & editing:** Michaela Kudrnáčová, Aleš Kudrnáč.

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
