## [Decision Letter · Decision Letter 0]

26 Sep 2022

PONE-D-22-14161Better Sleep, Better Life? Testing the Role of Sleep in Quality of LifePLOS ONE

Dear Dr. Kudrnáčová,

Thank you for submitting your manuscript to PLOS ONE. After careful consideration, we feel that it has merit but does not fully meet PLOS ONE’s publication criteria as it currently stands. Therefore, we invite you to submit a revised version of the manuscript that addresses the points raised during the review process.

We look forward to receiving your revised manuscript.

Kind regards,

Fakir Md Yunus, PhD, MSC, MPH, MBBS

Academic Editor

PLOS ONE

2.In your Data Availability statement, you have not specified where the minimal data set underlying the results described in your manuscript can be found. PLOS defines a study's minimal data set as the underlying data used to reach the conclusions drawn in the manuscript and any additional data required to replicate the reported study findings in their entirety. All PLOS journals require that the minimal data set be made fully available. For more information about our data policy, please see http://journals.plos.org/plosone/s/data-availability.

3.We note that you have indicated that data from this study are available upon request. PLOS only allows data to be available upon request if there are legal or ethical restrictions on sharing data publicly. For more information on unacceptable data access restrictions, please see http://journals.plos.org/plosone/s/data-availability#loc-unacceptable-data-access-restrictions.

4.Please include your full ethics statement in the ‘Methods’ section of your manuscript file. In your statement, please include the full name of the IRB or ethics committee who approved or waived your study, as well as whether or not you obtained informed written or verbal consent. If consent was waived for your study, please include this information in your statement as well.

Additional Editor Comments:

This is an interesting article and has merit for publication. However, there are several things needed to be cleared before it gets accepted. In addition to the reviewers’ comments, authors may consider to response to the following observations:

1. I don’t think this statement is correct “Sleep duration is a reliable predictor of wellbeing [26] and affects QoL. Kripke et al. [27] observed 1.1 million American adults for six years and found that both extremely short (less than 4 hours) and extremely long (more than 8 hours) periods of sleep lead to elevated mortality.” I don’t think more than 8 hours sleep is considered as extremely long period of sleep, I see that the reference is from 2002, 20 years earlier than today!! National sleep foundation recommended hours for 18-64 is 7-9 hours. https://www.sleepfoundation.org/how-sleep-works/how-much-sleep-do-we-really-need

2. “Higher sleep quality” and “improvements” are these two are two different thing? Same applies for “Higher social jetlag levels” and “increases in social jetlag”

3. “The average sleep duration in the CR is 7.5 hours (Table 1)” to me it information is not really helpful until it is presented by the age group.

4. Could you please consider explaining the H1 “Longer sleep duration and increases in sleep duration are related to higher levels of QoL.”. Firstly, please explain what do you mean by long sleep duration? How do you identify if someone is having longer explain? What is your cut-off? Secondly, “increases in sleep duration”, do you mean increase in sleep duration even if someone’s sleep duration fall within normal sleep hours. Please clarify.

5. I am not sure what study design authors used in this study. Is it a cross-section at multiple points of time or was this study a cohort study? Please explain why “the retention rate of individuals was 20.6 %” is important if this is not a cohort study.

6. I understand this was secondary data analysis? Authors may considering writing more about the dataset including the design used and cite articles that has been published using this dataset.

7. Please consider describing PAPI questionnaire? Could you please also some words on the why sample is higher in 2020 (2,161) than earlier year 2019 (2,046)? …..since you stated “the sample size in 2020 was reduced for financial reasons.”

8. What was the age of the adult population?

9. Authors mentioned that “4,523 respondents in up to 2,155 households.” So I was wondering who did the study interviewed? And how it was determined? Is it both male and female in the same household? Please kindly describe.

10. Regards to all measures, do the measures comprise of multiple items or single item measure? I see that wellness has multiple items and I’d appreciate if you can provide the composite Cronbach alpha. Plus provide if this measure is validated scale.

11. Also please consider mentioning the interpretation of the scale, i.e., what is the higher and lower score mean of some measures?

12. Authors did not mentioned about the normality assumptions of continuous variables? I see from table 1, max age is 94 years. I don’t know if it should be considered as outlier. Plus, SD is 22.51. Also the sleep duration is max 13.48 (hours I believe). I m sure if that should also be considered as outliers.

13. Presenting the table by data points (2018, 2019, and 2020) would be useful to see the mean changes over time. Seeing the wave 4-5-6 reduces the readability of the paper.

14. Table 1, please kindly explain the meaning of mean education 1.18. Is this the years of education completed? Also clarify the word “maturity exam” for increasing the readability for the international audience.

15. Please discuss the limitations of the item scale used all most of the measure since I believe it was measures by 1 item scoring. Plus I saw some scale asked specific time duration while some ae general subjective response without time boundness. For example, sleep duration and some other have not time restricted response while welling was over the last two weeks.

16. Table 2, please explain what does the numbers mean in the parenthesis. For example, 0.06 (−.14- .26)…..is this CI or SD or SEM. I suggest to write 95%CI.

17. What was your random effect variable in table 2 and table 3.

18. Table 2 and Table 3, please place the household income in USD for international audience. Or may be include both, local and international currencies.

19. I’d like to know more on how models were built. I see that all the independent variables were included in the final model. Also would you please provide the model fit statistics.

20. Could you please put the null model table as supplement.

Reviewers' comments:

Reviewer's Responses to Questions

**Comments to the Author**

1. Is the manuscript technically sound, and do the data support the conclusions?

Reviewer #1: Partly

Reviewer #2: Yes

2. Has the statistical analysis been performed appropriately and rigorously? 

Reviewer #1: Yes

Reviewer #2: Yes

3. Have the authors made all data underlying the findings in their manuscript fully available?

Reviewer #1: Yes

Reviewer #2: No

4. Is the manuscript presented in an intelligible fashion and written in standard English?

Reviewer #1: Yes

Reviewer #2: Yes

5. Review Comments to the Author

Reviewer #1: The manuscript Better Sleep, Better Life? Testing the Role of Sleep in Quality of Life has as main goal to establish the relationship between sleep or some of its features with quality of life. I consider that they have a good sample size, the questionnaire about QoL is very well chosen and explained.

However 1.the counterpart , the sleep questionnaires allow the participants to be very subjective, that could be observed in some questions that are al least inconsistent (as an example, the quality of sleep has a negative correlation with work stress in retired subjects, and there are other items alike).1. Sleep quality classified only by the answers without some complementary information is hard to interpret 3.Therefore some results that could be more objectives like sleep duration (time to sleep and wake up) or the social jet lag showed results similar to the literature and performed some of them with more more objective measures. 4. I think that to compare individuals and households did not give a particular contribution to the discussion. 5. We know that when the projects were designed , we did not know about the pandemic. But, the fact that the last year was an especial time of the life, it might not reflect the regular behavior of the participants and could introduce a bias to the final results. With this scenario to choose one year before (2016) for example would have been better.

It could be possible for increasing the interest of the research to create extreme groups (in sleep or QoL fields) and study if they have a different behavior.

In summary, the more important result that it appears to be that sleep quality is associated positively with QoL is based on a soft base that support similar results from other studies but does not contribute to deepen knowledge about the closest relationship between life asleep and life awake.

Reviewer #2: This study is imperative because it examined the impact of sleep on QoL. I would definitely recommend this paper for publication. The methodology of the study is solid and convincing, and the study findings enable further research on multidimensional QoL and sleeping patterns. I only have a few comments that could perhaps improve this write-up: 

1. I would like to suggest if the author could condense the introduction section. I find the introduction a bit lengthy and there is a mix of discussion in the introduction section. The summary of the previous research studies definitely helps readers to understand the background and loops of the previous studies. 

2. The authors highlighted the hypothesis of the study, but I would recommend a clear study objective in the introduction section.  

3. Some information in the method is unnecessary, i.e., "Case selection" could perhaps be written in the introduction section or maybe the discussion section, as the authors were comparing the Czech people with the other European nations. Or maybe they simplified the case selection of respondents that were included in this study. 

4. Please describe clearly 'PAPI' when first introduced. 

5. I would like to know exactly what tools were used to measure QoL in this study. Are the QoL tools newly developed? or adopted from other established QoL tools? Are the tools used specifically able to measure QoL for sleeping patterns or certain age groups? The authors need to elaborate on the tools used, especially on their suitability for certain age groups. older than adults? 6. Need more inputs on the discussion, especially comparing QoL and sleeping with other countries, authors' tools vs other QoL tools that measure QoL in sleeping.

6. Lastly, did the authors control for confounders such as those with mental health problems in the study or exclude those who are clinically diagnosed with sleeping problems in this study (if any).

I hope this helps. Thank you very much.

6. PLOS authors have the option to publish the peer review history of their article (what does this mean?). If published, this will include your full peer review and any attached files.

Reviewer #1: No

Reviewer #2: No

---

## [Author Response · Author response to Decision Letter 0]

24 Oct 2022

We thank the editors and the reviewers for the kind words and thorough comments and for the opportunity to R&R. The responses to reviewers are enclosed in the "Response to reviewers document".

---

## [Decision Letter · Decision Letter 1]

29 Nov 2022

PONE-D-22-14161R1Better Sleep, Better Life? Testing the Role of Sleep in Quality of LifePLOS ONE

Dear Dr. Kudrnáčová,

Thank you for submitting your manuscript to PLOS ONE. After careful consideration, we feel that it has merit but does not fully meet PLOS ONE’s publication criteria as it currently stands. Therefore, we invite you to submit a revised version of the manuscript that addresses the points raised during the review process. Please submit your revised manuscript by Jan 13 2023 11:59PM. If you will need more time than this to complete your revisions, please reply to this message or contact the journal office at plosone@plos.org. Please include the following items when submitting your revised manuscript:A rebuttal letter that responds to each point raised by the academic editor and reviewer(s). You should upload this letter as a separate file labeled 'Response to Reviewers'.A marked-up copy of your manuscript that highlights changes made to the original version. You should upload this as a separate file labeled 'Revised Manuscript with Track Changes'.An unmarked version of your revised paper without tracked changes. You should upload this as a separate file labeled 'Manuscript'.If applicable, we recommend that you deposit your laboratory protocols in protocols.io to enhance the reproducibility of your results. Protocols.io assigns your protocol its own identifier (DOI) so that it can be cited independently in the future. For instructions see: https://journals.plos.org/plosone/s/submission-guidelines#loc-laboratory-protocols. Additionally, PLOS ONE offers an option for publishing peer-reviewed Lab Protocol articles, which describe protocols hosted on protocols.io. Read more information on sharing protocols at https://plos.org/protocols?utm_medium=editorial-email&utm_source=authorletters&utm_campaign=protocols.

We look forward to receiving your revised manuscript.

Kind regards,

Fakir Md Yunus, PhD, MSC, MPH, MBBS

Academic Editor

PLOS ONE

Journal Requirements:

Additional Editor Comments:

Please kindly respond to Reviewer-1 comments.

"I would suggest to the authors:

1. Improve the quality of the table. For example , is better to do different tables for socio-economic variables, and other tables for the variables of the study. Besides , is cleaner if the intraindividual and inter-individual data could be seen apart.

It is important to reinforce that the study was done during COVID pandemia and might not be representative of the behavior in normal conditions."

Thank you.

Reviewers' comments:

Reviewer's Responses to Questions

**Comments to the Author**

1. If the authors have adequately addressed your comments raised in a previous round of review and you feel that this manuscript is now acceptable for publication, you may indicate that here to bypass the “Comments to the Author” section, enter your conflict of interest statement in the “Confidential to Editor” section, and submit your "Accept" recommendation.

Reviewer #1: All comments have been addressed

2. Is the manuscript technically sound, and do the data support the conclusions?

Reviewer #1: Yes

3. Has the statistical analysis been performed appropriately and rigorously? 

Reviewer #1: Yes

4. Have the authors made all data underlying the findings in their manuscript fully available?

Reviewer #1: Yes

5. Is the manuscript presented in an intelligible fashion and written in standard English?

Reviewer #1: Yes

6. Review Comments to the Author

Reviewer #1: I would suggest to the authors:

1. Improve the quality of the table. For example , is better to do different tables for socio-economic variables, and other tables for the variables of the study. Besides , is cleaner if the intraindividual and inter-individual data could be seen apart.

It is important to reinforce that the study was done during COVID pandemia and might not be representative of the behavior in normal conditions.

7. PLOS authors have the option to publish the peer review history of their article (what does this mean?). If published, this will include your full peer review and any attached files.

Reviewer #1: **Yes: **Cecilia Algarin

While revising your submission, please upload your figure files to the Preflight Analysis and Conversion Engine (PACE) digital diagnostic tool, https://pacev2.apexcovantage.com/. PACE helps ensure that figures meet PLOS requirements. To use PACE, you must first register as a user. Registration is free. Then, login and navigate to the UPLOAD tab, where you will find detailed instructions on how to use the tool. If you encounter any issues or have any questions when using PACE, please email PLOS at figures@plos.org. Please note that Supporting Information files do not need this step.<quillbot-extension-portal></quillbot-extension-portal>

---

## [Author Response · Author response to Decision Letter 1]

2 Dec 2022

Response to Reviewers is uploaded in the "Attach Files" section.

---

## [Editor Report · Decision Letter 2]

8 Feb 2023

Better Sleep, Better Life? Testing the Role of Sleep in Quality of Life

PONE-D-22-14161R2

Dear Dr. Kudrnáčová,

We’re pleased to inform you that your manuscript has been judged scientifically suitable for publication and will be formally accepted for publication once it meets all outstanding technical requirements.

Kind regards,

Fakir Md Yunus, PhD, MSC, MPH, MBBS

Academic Editor

PLOS ONE

Additional Editor Comments (optional):

Reviewers' comments:

<quillbot-extension-portal></quillbot-extension-portal>

---

## [Editor Report · Acceptance letter]

22 Feb 2023

PONE-D-22-14161R2 

Better sleep, better life? Testing the role of sleep on quality of life 

Dear Dr. Kudrnáčová:

I'm pleased to inform you that your manuscript has been deemed suitable for publication in PLOS ONE. Congratulations! Your manuscript is now with our production department. 

Kind regards, 

on behalf of

Dr. Fakir Md Yunus 

Academic Editor

PLOS ONE